# The Utility of Ultrasound in Evaluating Joint Pain in Systemic Lupus Erythematosus: Looking beyond Fibromyalgia

**DOI:** 10.3390/jpm13050763

**Published:** 2023-04-28

**Authors:** Yeohan Song, Gabriel Kirsch, Wael Jarjour

**Affiliations:** 1Department of Internal Medicine, The Ohio State University Wexner Medical Center, Columbus, OH 43210, USA; 2Division of Rheumatology and Immunology, The Ohio State University Wexner Medical Center, Columbus, OH 43210, USA

**Keywords:** systemic lupus erythematosus, ultrasound, diagnostic, therapeutic, fibromyalgia, inflammatory arthritis, joint pain

## Abstract

Background: Systemic lupus erythematosus (SLE) is a complex autoimmune condition with varied clinical presentations, and musculoskeletal pain is one of the most commonly associated symptoms. However, fibromyalgia (FM) is a prevalent co-existing condition in SLE patients that can also cause widespread pain, and in patients with both conditions, it is often difficult to distinguish the underlying cause of musculoskeletal pain and provide optimal therapy. Methods: A retrospective cohort study was conducted including all adult SLE patients who received musculoskeletal ultrasound (US) examinations for joint pain at the Ohio State University Wexner Medical Center between 1 July 2012, and 30 June 2022. Binary and multiple logistic regression analyses were performed to determine predictors of US-detected inflammatory arthritis as well as improved musculoskeletal pain. Results: A total of 31 of 72 SLE patients (43.1%) had a co-existing diagnosis of FM. In binary logistic regression, a co-existing diagnosis of FM was not significantly associated with US-detected inflammatory arthritis. In multiple logistic regression analysis, clinically detected synovitis was significantly associated with US-detected inflammatory arthritis (aOR, 142.35, *p* < 0.01), and there was also a weak association with erythrocyte sedimentation rate (ESR) (aOR 1.04, *p* = 0.05). In separate multiple logistic regression analysis, US-guided intra-articular steroid injection was the only predictor of improved joint pain at follow-up visit (aOR 18.43, *p* < 0.001). Conclusions: Musculoskeletal US can be an effective modality to detect inflammatory arthritis as well as to guide targeted intra-articular steroid injection to alleviate joint pain in SLE patients with or without FM.

## 1. Introduction

Systemic lupus erythematosus (SLE) is a complex autoimmune condition characterized by autoantibodies and aberrant immune responses involving one or more organ systems that affects more than 3 million people worldwide [1]. Musculoskeletal pain is one of the most commonly reported symptoms associated with this condition, with over 80% of SLE patients experiencing joint pain—often as one of the earliest presenting symptoms [2,3]. While deforming arthropathies such as those seen in Jaccoud arthropathy or rhupus syndrome can occasionally be observed, joint involvement in SLE is typically characterized as an inflammatory arthralgia with non-erosive, non-deforming arthritis [4,5].

The heterogeneous clinical presentation of SLE has led various international working groups and organizations to construct classification criteria to improve characterization of affected patients for research studies, while also updating these criteria to incorporate the adoption of immunologic laboratory tests in clinical practice and to reflect the improved understanding of the observed patterns of organ involvement. Two of the most widely recognized among these, the 2019 European League Against Rheumatism (EULAR)/American College of Rheumatology (ACR) classification criteria and the 2012 Systemic Lupus International Collaborating Clinics (SLICC) classification criteria, both include measures of joint tenderness among the clinical criteria for SLE [2,6]. However, it is notable that clinically detected joint inflammation had one of the lowest specificities for SLE (43.6%) in the derivation sample for the SLICC classification criteria, suggesting that there can be other potential etiologies of joint pain among these patients [6].

Among these other causes of musculoskeletal pain, fibromyalgia (FM) is a condition characterized by chronic and widespread pain that affects 1–2% of the general adult population [7,8], with an even higher prevalence observed in SLE patients (estimates range from 6% to greater than 20%) [9,10]. Though the etiology of FM remains to be fully elucidated and may be multifactorial, as a chronic pain syndrome, neurochemical studies suggest that the pathophysiology underlying FM involves abnormal pain modulation and central sensitization [11,12,13]. As such, co-existing FM may complicate accurate clinical assessment of musculoskeletal pain in SLE patients. Additionally, with the availability of newer immunosuppressive agents and more robust therapeutic regimens, the life expectancy of SLE patients has increased [14,15] and an increasing number are experiencing osteoarthritis, which represents another potential etiology of musculoskeletal pain [16]. Furthermore, patients with both FM and osteoarthritis have been shown to experience higher perceived joint pain than patients with either condition alone [17], which may also affect the accuracy of clinical joint assessment for synovial inflammation, or synovitis, in these patients.

It has previously been reported that SLE patients without co-existing FM who have sonographic features of inflammatory arthritis in the hands and wrists are more likely to respond to systemic glucocorticoid therapy [18]. While this can provide guidance for adjustments to systemic therapies, in those SLE patients who report joint pain localized to a specific joint and/or also have co-existing FM, it remains to be determined whether a targeted musculoskeletal ultrasound (US) examination accompanied by intra-articular steroid injection can also provide therapeutic benefit and improvement in joint pain, as intra-articular steroid injections can effectively alleviate joint pain from a range of causes, including osteoarthritis [19,20,21], rheumatoid arthritis [22], adhesive capsulitis [23], and potentially crystalline arthropathy [24,25]. In recognition of this, organizations such as the ACR as well as the American Academy of Orthopaedic Surgeons (AAOS) recommend intra-articular steroid injection as a treatment for osteoarthritis and associated joint pain [20,21].

Musculoskeletal US examination of the joints allows further diagnostic examination of the synovial membrane and intra-articular space as well as adjacent bone cortices to assess for the presence of synovitis and erosive changes associated with inflammatory arthritis from any cause, including rheumatoid arthritis and psoriatic arthritis as well as SLE. The sonographic features associated with inflammatory arthritis are well described in consensus definitions from the Outcome Measures in Rheumatoid Arthritis Clinical Trials (OMERACT) US special interest group and include gray-scale changes detected in B-mode indicating synovial hypertrophy (abnormal hypoechoic, non-displaceable, and poorly compressible intra-articular tissue); joint effusion (increased hypoechoic or anechoic, displaceable, and compressible intra-articular material); tenosynovitis (hypoechoic or anechoic thickened tissue in the tendon sheath observed in orthogonal planes); and bone erosion (intra-articular discontinuity of bone cortices observed in orthogonal planes) [26]. Power Doppler (PD) US has also been used to detect synovial vascularization associated with synovitis [27,28].

Particularly among SLE patients without detectable radiographic changes associated with arthritis, such as bone erosions, previous studies have demonstrated that musculoskeletal US can be a useful tool to detect subclinical synovitis, or early joint inflammation, as well as to confirm the presence of clinically detected synovitis [27,28,29,30,31]. Despite this agreement on the value of musculoskeletal US for this indication, these studies have had varying reports on the significance of associations between sonographic features of inflammatory arthritis in SLE and laboratory tests routinely obtained in clinical practice in the assessment of joint inflammation, such as the erythrocyte sedimentation rate (ESR) and C-reactive protein (CRP), as well as with those that can fluctuate with SLE disease activity, such as anti-double-stranded DNA (anti-dsDNA) and complement component 3 (C3) and complement component 4 (C4) levels, suggesting that these laboratory markers alone are insufficient to determine the presence of subclinical or clinical synovitis [27,31,32]. As SLE patients often have additional conditions associated with musculoskeletal pain, such as FM and osteoarthritis, musculoskeletal US has further potential diagnostic and therapeutic utility in distinguishing inflammatory arthritis from other causes of musculoskeletal pain and in the accurate delivery of targeted intra-articular injections to alleviate joint pain in these patients, which is explored in this study.

## 2. Materials and Methods

Study design. A retrospective cohort study was conducted including all adult patients (i.e., ≥18 years of age) with a documented diagnosis of SLE from a rheumatologist who received musculoskeletal US examinations for joint pain in an outpatient setting at The Ohio State University Wexner Medical Center between 1 July 2012, and 30 June 2022. Of note, musculoskeletal US examinations were performed in a dedicated clinic by a rheumatologist or sports medicine physician with privileges to perform US-guided arthrocentesis at the Medical Center (US practitioner) separately from the patient’s routine clinic visits with his or her managing rheumatologist. Musculoskeletal US practitioners were required to complete a minimum of 16 h of musculoskeletal US-specific continuing medical education (CME) including didactics and hands-on training, as well as a minimum of 25 supervised US examinations. Musculoskeletal US examinations and intra-articular steroid injections, when performed, were targeted to the single joint where the patient reported the greatest pain. If a patient was examined more than once in the study period, the first US examination was included to maintain independence of observations within the study sample. Exclusion criteria included the following: patients < 18 years of age, those who did not have a diagnosis of SLE prior to the time of musculoskeletal US examination, and those with prior surgical instrumentation of the joint examined by US. Patient medical records fulfilling the above criteria were requested through The Ohio State University Wexner Medical Center Information Warehouse, which were then manually reviewed to confirm study eligibility. Study data were recorded in a de-identified manner to protect patient confidentiality.

Clinical and Laboratory Data. Medical records of eligible patients were reviewed, and demographic (e.g., age and body mass index [BMI] on date of musculoskeletal US, sex, and race) data were recorded. Documentation produced by the patient’s managing rheumatologist preceding the musculoskeletal US examination was then reviewed to determine if there was clinically detected inflammatory arthritis (presence of joint swelling/effusion and/or joint tenderness with ≥30 min of morning stiffness, based on the 2019 EULAR/ACR and 2012 SLICC classification criteria) [2,6], as well as whether the patient had a co-existing diagnosis of FM based on the 2010 ACR diagnostic criteria [33]. Laboratory parameters including anti-dsDNA, C3, C4, ESR, and CRP levels preceding the US examination were then recorded. Based on the procedure note on the date of the musculoskeletal US examination, the presence or absence of US-detected inflammatory arthritis was then recorded as determined by the US practitioner’s evaluation of synovial hypertrophy, joint effusion, tenosynovitis, and bone erosion (Figure 1 and Figure 2). If a joint aspiration and/or intra-articular steroid injection was performed during the musculoskeletal US examination, this was noted. If systemic therapy such as oral or intra-muscular steroids or immunosuppression was started or increased based on the US findings, this was recorded. Finally, the number of days leading up to the follow-up clinic visit with the patient’s managing rheumatologist for re-assessment of joint pain after the musculoskeletal US examination, as well as whether the patient reported improvement in joint pain at the follow-up visit, were also recorded.

Data Analyses. Data were analyzed using SPSS version 28 (IBM Corp. Released 2021. IBM SPSS Statistics for Windows, Version 28.0. IBM Corp.: Armonk, NY, USA). Descriptive statistics summaries including frequencies and cross-tabulations were used to characterize the data. Correlation matrices were reviewed for statistically significant (α = 0.05, two-tailed tests) associations among potential predictor variables, and significantly correlated variables were excluded to avoid multicollinearity. Binary logistic regression analysis was then used to identify predictor variables associated with the outcome variable (*p*-value ≤ 0.20), which were then fitted into multiple logistic regression models using the backward elimination method. Predictor variables with *p*-values less than 0.05 in multiple logistic regression were considered statistically significant, and the strengths of association were interpreted using the adjusted odds ratio (aOR) and 95% confidence interval (CI).

## 3. Results

### 3.1. Patient Characteristics

A total of 72 adult SLE patients (66 female, 6 male) received musculoskeletal US examinations for joint pain in the study period, with a median age of 50 years (interquartile range [IQR]: 40–63 years). Of these patients, 31 (43.1%) had a co-existing diagnosis of FM preceding the musculoskeletal US. The patients self-reported their race as the following: 30 (41.7%) were White/Caucasian, 33 (45.8%) were Black/African American, 4 (5.6%) were Asian American, and 4 (5.6%) were Other, with 1 (1.4%) declining to answer. The mean BMI was 32.3 kg/m^2^ (±SD 9.4 kg/m^2^). Prior to the musculoskeletal US examination, 6 (8.3%) of the patients were found to have clinically detected synovitis based on the previously described criteria. A summary of the demographic characteristics of SLE patients who had or did not have a co-existing diagnosis of FM is shown in Table 1.

Nearly all SLE patients were on hydroxychloroquine prior to the musculoskeletal US examination, including 29 (93.5%) of the patients with co-existing FM and 37 (90.2%) of those without co-existing FM. Those who were not on hydroxychloroquine were noted to have previously discontinued this medication due to side effects. Additionally, a greater proportion of patients with co-existing FM were on systemic steroids prior to the US examination (45.2% vs. 36.6%; prednisone dose range: 2.5 mg to 20 mg daily), while relatively more patients without co-existing FM were on steroid-sparing immunosuppression (51.2% vs. 32.3%), with mycophenolate and methotrexate being the most common immunosuppressants used (45.2% and 29.0% of all immunosuppression, respectively).

### 3.2. Laboratory Parameters

Laboratory measures for this cohort of SLE patients with joint pain prior to the US examination included (medians and IQRs): anti-dsDNA (4.0 IU/mL, 4.0–12.0 IU/mL), C3 (131.0 mg/dL, 114.0–144.0 mg/dL), C4 (30.0 mg/dL, 19.0–37.0 mg/dL), ESR (23.5 mm/h, 11.8–49.5 mm/h), and CRP (3.8 mg/L, 1.3–10.3 mg/L). Stratification of these laboratory measures according to whether the patient had FM is presented in Table 2. Among the 31 SLE patients with co-existing FM, 15 (48.4%) had either ESR above the laboratory reference range of 30.0 mm/h or CRP above the reference range of 10.0 mg/L, whereas 18 (43.9%) of the 41 SLE patients without co-existing FM had ESR or CRP above the reference range. Among these patients with either ESR or CRP above the reference range, 21 (63.6%) also had a positive anti-ribonucleoprotein (anti-RNP) antibody, whereas 8 (24.2%) had negative anti-RNP and 4 (12.1%) did not have this tested.

### 3.3. Ultrasound Examination

Musculoskeletal US examinations were most often performed on a knee joint (20 patients, 27.8%). Twelve patients (16.7%) each had an US examination of a shoulder or hip joint. Eight patients (11.1%) had an US examination of a joint of the hand. Five patients (6.9%) each had an US examination of a wrist or ankle joint. The remaining patients had an US examination of an elbow joint (4 patients, 5.6%) or a foot or sacroiliac joint (3 patients or 4.2% each).

On musculoskeletal US examination, the US practitioners detected the presence of inflammatory arthritis in 8 patients, 5 (62.5%) of whom had synovitis detected on clinical examination. The reported sonographic features of inflammatory arthritis included gray-scale changes indicating both synovial hypertrophy and joint effusion in 7 (87.5%) of these patients and tenosynovitis in 2 patients (25.0%), with PD signal detected in the joints of 3 patients (37.5%). Bone erosions were not detected in any of these patients. Of these SLE patients with US-detected inflammatory arthritis, 3 (37.5%) also had a diagnosis of FM.

During the musculoskeletal US examination, 58 patients (80.6%) received an intra-articular steroid injection, and 9 patients (12.5%) had an aspiration of a joint effusion. Eight patients (11.1%) received additional systemic steroids and/or immunosuppression from their managing rheumatologist upon receiving the US examination report. The median time to follow-up visit with the managing rheumatologist after the US examination was 77 days (IQR: 44–132.5 days).

### 3.4. Statistical Analysis

Descriptive statistics summaries revealed significant correlations among several potential predictor variables. A significant correlation between BMI and both ESR (*r* = 0.34, *p* < 0.01), and CRP (*r* = 0.43, *p* < 0.001) was found, as has been previously described in patients with inflammatory arthritis [34]. As ESR and BMI were less significantly associated than CRP and BMI, as also described by other groups [34], BMI and CRP were excluded and ESR was retained to avoid multicollinearity in regression analysis. Additional correlations between C3 and C4 (*r* = 0.61, *p* < 0.001), ESR (*r* = 0.28, *p* < 0.05), and CRP (*r* = 0.45, *p* < 0.001) were noted; therefore, C3 was also excluded and C4 was retained for further regression analysis. Similarly, race was also excluded, as it was significantly associated with ESR based on analysis of variance (*F* = 7.88, *p* < 0.001).

On binary logistic regression, age, sex, and co-existing diagnosis of FM were not found to be associated with US-detected inflammatory arthritis (Table 3). Multiple logistic regression analysis of the remaining predictors, including anti-dsDNA, C4, ESR, and clinically detected synovitis, showed clinically detected synovitis had the most significant association with and the highest odds of US-detected inflammatory arthritis in the final model (aOR = 142.35, 95% confidence interval [CI] 6.55–3093.96; *p* = 0.002), as well as a borderline significant association with ESR levels (aOR = 1.04, 95% CI 1.00–1.08; *p* = 0.05).

The results of additional binary logistic regression analyses to identify significant predictors of reported improvement in joint pain at the follow-up visit with the managing rheumatologist after musculoskeletal US examination are shown in Table 4. Age, sex, anti-dsDNA, C4, ESR, clinically detected synovitis, and joint aspiration were not found to be associated with improved joint pain after US examination. Multiple logistic regression analysis of the remaining predictors, including co-existing diagnosis of FM, intra-articular steroid injection, and additional systemic steroids and/or immunosuppression based on US findings, showed intra-articular steroid injection was the only predictor significantly associated with improved joint pain at follow-up visit (aOR = 18.43, 95% CI 3.67–92.55; *p* < 0.001).

## 4. Discussion

As reported in the United States and globally, SLE patients are predominantly (>85%) female [1,35], which was reflected in the patient sample in this study. Consistent with another study on SLE patients with musculoskeletal pain, the proportion of patients diagnosed with FM in this study was higher than that observed among unselected SLE patient cohorts [18]. In agreement with previous reports on the use of musculoskeletal US in the detection of subclinical and clinical synovitis in SLE patients, anti-dsDNA and complement levels were not found to be significantly associated with the presence of sonographic features of synovitis [31,32].

The results of this study support the association of the sonographic features of inflammatory arthritis with the clinical assessment of synovitis, which remains a well-recognized parameter in both the clinical diagnosis and classification criteria of SLE [2,6]. Care should be taken not to allow a preceding diagnosis of FM to affect the determination of whether inflammatory arthritis is present, particularly among patients with SLE, who have a higher prevalence of co-existing FM [9,10]. This study demonstrates the additional benefit musculoskeletal US can provide for these patients, as particularly among SLE patients who may have multiple causes of pain, diagnostic US can be an effective non-invasive method of corroborating suspected inflammatory arthritis based on the provided history and examination. Additionally, although clinicians should not be overly reliant on laboratory tests and should correlate abnormal findings with the examination, as shown in this study, elevated ESR levels can also potentially provide support for the presence of inflammatory arthritis among SLE patients with joint pain when considered in the appropriate clinical context.

Musculoskeletal US as an imaging modality to improve detection of pre-clinical synovitis in SLE has been well described [27,28,29,30,31]. It has also been reported that in SLE patients without FM, the presence of sonographic features of inflammatory arthritis may help stratify those patients who may benefit from adjustments to systemic therapy, such as intramuscular steroids [18]. However, as demonstrated in this study, there may be additional utility in the targeted use of musculoskeletal US to differentiate inflammatory arthritis from other co-existing causes of musculoskeletal pain in SLE patients, including FM and/or osteoarthritis, and to provide targeted therapeutic intervention in the form of an image-guided intra-articular steroid injection during the examination. This is of particular importance in SLE, which has a well-documented risk of complications such as osteoporotic fractures, avascular necrosis, coronary artery disease, and cataracts that increases with cumulative systemic steroid exposure [36,37].

Among SLE patients with joint pain, use of musculoskeletal US has the additional benefit of providing image guidance for intra-articular steroid injections, which can ensure accurate medication delivery based on real-time anatomic visualization. The targeted delivery of an intra-articular injection into a specific joint has the additional appeal of lower or less rapid systemic absorption compared to oral, intramuscular, or intravenous formulations, as peak serum steroid levels appear to be affected more by the number of joints injected rather than by the dose alone, though individual variability can be observed [38,39]. Additionally, while there has been concern regarding potential risk for osteoarthritis progression, avascular necrosis, and articular collapse following intra-articular steroid injections, a recent study did not demonstrate this association after controlling for baseline osteoarthritis severity and pre-existing avascular necrosis/subchondral insufficiency fracture [40]. However, caution is advised and patient-specific risks should be considered in patients at greater risk of adverse outcomes, such as those who have diabetes mellitus and therefore can experience higher post-injection glycemic levels [41], as well as those already on systemic steroids [38], and those who are anticipating arthroplasty within 3 months, as this increases the risk of periprosthetic joint infection [42]. Furthermore, intra-articular injections should be avoided entirely in patients who have clinical evidence of skin or soft tissue infection overlying the target joint.

In cases where joint pain reported by SLE patients is localized to a specific joint, the presented findings suggest that there may be an additional role for more targeted US assessments of the joints involved, which can provide confirmatory findings of inflammatory arthritis in cases complicated by multiple other potential causes of joint pain and inform decisions on adjustments to systemic therapies. Moreover, during the US examination, accurate image-guided administration of intra-articular steroid medications can provide therapeutic benefit to alleviate the pain experienced by these patients. Furthermore, SLE patients with FM who do not have US-detected features of inflammatory arthritis may also benefit from intra-articular steroid injections to improve joint pain from other causes, as these patients have been shown to have higher perceived pain from other joint pathology such as osteoarthritis [17].

Limitations of this study include the retrospective design of the study and that all patients were treated at a single academic healthcare institution. Due to the retrospective nature of the study, the presence or absence of US-detected inflammatory arthritis was recorded as binary data from the text of the musculoskeletal US examination notes, as the US practitioners did not document standardized scores and for the patients who received intra-articular steroid injections, the corresponding pre-injection US images were not consistently available for review. Outcomes such as patient-reported improvement in musculoskeletal pain were similarly limited to binary data based on the medical documentation. The median follow-up in this study was between 2 and 3 months, which aligns with the reported duration of pain alleviation following intra-articular steroid injections for conditions such as osteoarthritis [21]. Future prospective studies may further characterize the degree of joint inflammation on US using a standardized scoring system and the extent and duration of improvement in pain using scoring systems such as the visual analogue scale (VAS) over a longer follow-up period. Additionally, the minimum institutional training requirement for musculoskeletal US practitioners as detailed in the text may limit the applicability of the findings to other institutions that do not have similar requirements. Future prospective studies may also benefit from a single dedicated US practitioner to examine all recruited subjects to eliminate inter-observer variability. All patients in this study were ≥18 years of age, and findings may not be applicable to younger patient groups. None of the patients in this study had prior surgical instrumentation of the joint examined by musculoskeletal US, as management of post-operative joint pain would be deferred to the operating orthopedist.

Furthermore, as a lack of association between conventional SLE disease activity indices such as the Systemic Lupus Erythematosus Disease Activity Index 2000 (SLEDAI-2K) and the European Consensus Lupus Activity Measurement (ECLAM) and sonographic features of inflammatory arthritis has previously been demonstrated, these indices were not included in this study [31]. It has been suggested that these indices are not designed to identify isolated local joint inflammation in the absence of systemic or major organ involvement, indicating the need for an assessment that incorporates imaging modalities, such as musculoskeletal US, to improve detection of subclinical synovitis [31].

## 5. Conclusions

Musculoskeletal US can be an effective modality to distinguish inflammatory arthritis from other causes of musculoskeletal pain in SLE patients, as well as to guide targeted intra-articular steroid injection for the alleviation of joint pain, regardless of whether these patients have a co-existing diagnosis of FM.

## Figures and Tables

**Figure 1 jpm-13-00763-f001:**
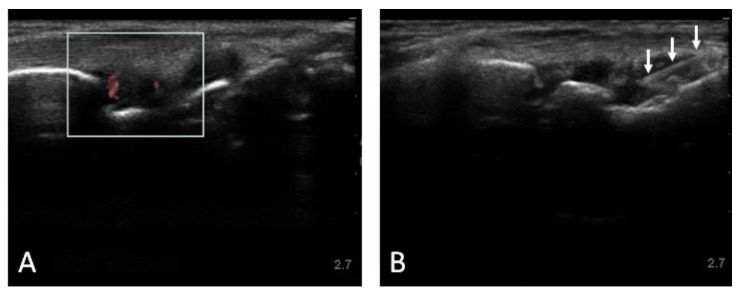
Musculoskeletal US images of SLE patient without FM reporting right wrist pain, with sonographic features of inflammatory arthritis. (**A**) Dorsal longitudinal view of the right wrist showing joint effusion with associated power Doppler signal; (**B**) Dorsal longitudinal view of the right wrist during intra-articular injection, with the needle entering the joint space from the 2 o’clock position (*arrows*). Depth in centimeters shown on the right of each image.

**Figure 2 jpm-13-00763-f002:**
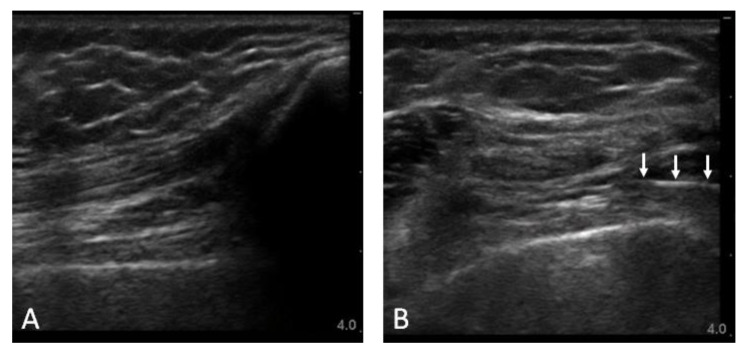
Musculoskeletal US images of SLE patient with FM reporting left knee pain, without sonographic features of inflammatory arthritis. (**A**) Anterior longitudinal view of the left knee showing the non-distended suprapatellar bursa; (**B**) Anterior transverse view of the left knee during intra-articular injection, with the needle entering the suprapatellar bursa from the 3 o’clock position (*arrows*) and slightly distending the suprapatellar bursa with injectate. Depth in centimeters shown on the right of each image.

**Table 1 jpm-13-00763-t001:** Demographic characteristics of SLE patients with joint pain stratified by co-existing diagnosis of FM.

	SLE with FM (*n* = 31)	SLE without FM (*n* = 41)
Age ^1^ (years, median [IQR])	48 (40.5–61)	50 (40–63)
Sex		
Male (*n*, %)	1 (3.2)	5 (12.2)
Female (*n*, %)	30 (96.8)	36 (87.8)
Race ^2^		
White (*n*, %)	18 (58.1)	12 (29.3)
Black (*n*, %)	11 (35.5)	22 (53.7)
Asian (*n*, %)	1 (3.2)	3 (7.3)
Other (*n*, %)	1 (3.2)	3 (7.3)
BMI (kg/m^2^, mean ± SD)	33.9 ± 10.0	31.1 ± 8.9

^1^ Age on date of musculoskeletal US examination. ^2^ One SLE patient without FM declined to self-report race. BMI body mass index, FM fibromyalgia, SLE systemic lupus erythematosus.

**Table 2 jpm-13-00763-t002:** Laboratory parameters of SLE patients with joint pain stratified by co-existing diagnosis of FM.

	SLE with FM (*n* = 31)	SLE without FM (*n* = 41)
anti-dsDNA (IU/mL)	4.0 (4.0–9.5)	4.5 (4.0–22.5)
C3 (mg/dL)	132.0 (104.0–151.0)	130.0 (117.8–140.8)
C4 (mg/dL)	28.0 (18.0–33.0)	32.0 (25.8–37.3)
ESR (mm/h)	23.5 (10.8–44.0)	24.5 (12.0–61.5)
CRP (mg/L)	2.9 (1.6–9.5)	4.3 (1.3–11.1)

Data shown as median, IQR. Anti-dsDNA anti-double-stranded DNA, C3 complement component 3, C4 complement component 4, CRP C-reactive protein, ESR erythrocyte sedimentation rate, FM fibromyalgia, SLE systemic lupus erythematosus.

**Table 3 jpm-13-00763-t003:** Binary logistic regression of predictor variables for inflammatory arthritis detected on musculoskeletal US examination of SLE patients with joint pain.

	Odds Ratio (95% CI)	*p*-Value
Age	1.03 (0.97, 1.08)	0.35
Sex	0.59 (0.06, 5.83)	0.65
anti-dsDNA	1.01 (1.00, 1.03)	0.06 *
C4	0.91 (0.83, 0.99)	0.03 *
ESR	1.02 (0.99, 1.04)	0.16 *
Clinical synovitis	105.00 (9.16, 1204.08)	<0.001 *
Co-existing FM	0.77 (0.17, 3.51)	0.74

* Predictor variables associated with the outcome variable (*p*-value ≤ 0.20) were included in multiple logistic regression analysis. Anti-dsDNA anti-double-stranded DNA, C4 complement component 4, ESR erythrocyte sedimentation rate, FM fibromyalgia, SLE systemic lupus erythematosus, US ultrasound.

**Table 4 jpm-13-00763-t004:** Binary logistic regression of predictor variables for patient reported improvement in pain at follow-up visit after musculoskeletal US examination of SLE patients with joint pain.

	Odds Ratio (95% CI)	*p*-Value
Age	1.00 (0.97, 1.04)	0.95
Sex	1.83 (0.34, 9.79)	0.48
anti-dsDNA	0.99 (0.98, 1.00)	0.22
C4	1.00 (0.96, 1.03)	0.86
ESR	1.00 (0.99, 1.02)	0.63
Clinical synovitis	0.55 (0.10, 2.94)	0.48
Co-existing FM	2.84 (1.00, 8.07)	0.05 *
Intra-articular steroid injection	18.43 (3.67, 92.55)	<0.001 *
Joint aspiration	0.69 (0.17, 2.83)	0.60
Additional systemic therapy ^1^	0.30 (0.07, 1.38)	0.12 *

^1^ Oral or intra-muscular steroids and/or immunosuppression started or increased based on musculoskeletal US examination findings. * Predictor variables associated with the outcome variable (*p*-value ≤ 0.20) were included in multiple logistic regression analysis. Anti-dsDNA anti-double-stranded DNA, C4 complement component 4, ESR erythrocyte sedimentation rate, FM fibromyalgia, SLE systemic lupus erythematosus, US ultrasound.

## Data Availability

Requests for data presented in this study should be submitted to the Department of Medical Information Management at The Ohio State University Wexner Medical Center, which controls access to its patient information used for research purposes.

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
