# Peer review of "The Utility of Ultrasound in Evaluating Joint Pain in Systemic Lupus Erythematosus: Looking beyond Fibromyalgia"

_jpm, 2023, doi:10.3390/jpm13050763_

Round 1

Reviewer 1 Report

Interesting, well- designed study, trying to elucidate a common diagnostic problem in everyday clinical practice. Generally, US joint examination is helpful in all patients with inflammatory arthritis ( including seronegative Rheumatoid arthritis that may also frequently present without erosions in x rays), not only in SLE. Since this study focuses on SLE particularly, SLEDAI scores could be reported, in order to better indicate disease activity, and therefore further statistical analysis could be performed. 

 Moreover, the ultrasound examination is performed by several different practicioners in a ten-year period. Since this kind of examination is subjective, the results may widely vary. In such cases, it would be wiser to use a specific US practitioner, in order to examine all the recruited subjects. However, it is clearly stated that all practitioners were equally trained.

The use of English language is excellent. The manuscript is well written and adequately understood.

Reviewer 2 Report

The article is interesting and relevant. A very long introduction, it can be shortened. It is often difficult to assess the presence of inflammation in SLE, especially after the age of 40. The authors in the materials and methods indicate an average age of 50 years, it is better to specify the median age and 25%-75% variation. In addition, the average level of DRR is 5.5, it is also better to give a median and spreads. It is also necessary to indicate the number of patients with elevated CRP and ESR, whether they have been studied for RNP, these patients with SLE often have arthritis and increased acute phase indices such as CRP, ESR, fibrinigen. It would be good to reflect therapy in groups of patients in the materials and methods and indicate the criteria for the diagnosis of fibromyalgia.

Reviewer 3 Report

The article is well done, but in the last years several studies highlight the value of US in SLE.

Regarding the title "Utility of US in improving joint pain", I think it would be better "Utility of US in evaluating joint pain".
